# Changes in Left Heart Geometry, Function, and Blood Serum Biomarkers in Patients with Obstructive Sleep Apnea after Treatment with Continuous Positive Airway Pressure

**DOI:** 10.3390/medicina58111511

**Published:** 2022-10-24

**Authors:** Laima Kondratavičienė, Eglė Tamulėnaitė, Eglė Vasylė, Andrius Januškevičius, Eglė Ereminienė, Kęstutis Malakauskas, Marius Žemaitis, Skaidrius Miliauskas

**Affiliations:** 1Department of Pulmonology, Lithuanian University of Health Sciences, 44307 Kaunas, Lithuania; 2Department of Cardiology, Lithuanian University of Health Sciences, 44307 Kaunas, Lithuania; 3Laboratory of Pulmonology, Department of Pulmonology, Lithuanian University of Health Sciences, 44307 Kaunas, Lithuania

**Keywords:** obstructive sleep apnea, continuous positive airway pressure, two-dimensional speckle-tracking echocardiography, galectin-3, sST2, endothelin-1

## Abstract

*Background:* Cardiovascular remodeling is essential in patients with obstructive sleep apnea (OSA), and continuous positive airway pressure (CPAP) therapy could improve these processes. Two-dimensional (2D) speckle-tracking (ST) echocardiography is a useful method for subclinical biventricular dysfunction diagnosis and thus might help as an earlier treatment for OSA patients. It is still not clear which blood serum biomarkers could be used to assess CPAP treatment efficacy. *Objectives:* To evaluate left heart geometry, function, deformation parameters, and blood serum biomarker (galectin-3, sST2, endothelin-1) levels in patients with OSA, as well as to assess changes after short-term CPAP treatment. *Materials and Methods:* Thirty-four patients diagnosed with moderate or severe OSA, as well as thirteen patients as a control group, were included in the study. All the subjects were obese (body mass index (BMI) > 30 kg/m^2^). Transthoracic 2D ST echocardiography was performed before and after 3 months of treatment with CPAP; for the control group, at baseline only. Peripheral blood samples for the testing of biomarkers were collected at the time of study enrolment before the initiation of CPAP therapy and after 3 months of CPAP treatment (blood samples were taken just for OSA group patients). *Results:* The left ventricle (LV) end-diastolic diameter and volume, as well as LV ejection fraction (EF), did not differ between groups, but an increased LV end-systolic volume and a reduced LV global longitudinal strain (GLS) were found in the OSA group patients (*p* = 0.015 and *p* = 0.035, respectively). Indexed by height, higher LV MMi in OSA patients (*p* = 0.007) and a higher prevalence of LV diastolic dysfunction (*p* = 0.023) were found in this group of patients. Although left atrium (LA) volume did not differ between groups, OSA group patients had significantly lower LA reservoir strain (*p* < 0.001). Conventional RV longitudinal and global function parameters (S′, fractional area change (FAC)) did not differ between groups; however, RV GLS was reduced in OSA patients (*p* = 0.026). OSA patients had a significantly higher right atrium (RA) diameter and mean pulmonary artery pressure (PAP) (*p* < 0.05). Galectin-3 and sST2 concentrations significantly decreased after 3 months of CPAP treatment. *Conclusions:* OSA is associated with the left heart remodeling process—increased LV myocardial mass index, LV diastolic dysfunction, reduced LV and RV longitudinal strain, and reduced LA reservoir function. A short-term, 3-months CPAP treatment improves LV global longitudinal strain and LA reservoir function and positively affects blood serum biomarkers. This new indexing system for LV myocardial mass by height helps to identify myocardial structural changes in obese patients with OSA.

## 1. Introduction

Obstructive sleep apnea (OSA) is described as a chronic disease with repeatable respiratory pauses during sleep, followed by episodic hypoxia and sleep fragmentation [1]. Upper airway obstruction is associated with significantly decreased intrathoracic pressure at inspiration, which leads to the increased return of venous blood to the heart and increased wall stress on both the atrium and ventricles [2]. Hypoxia has the potential to modulate the tissue remodeling processes or the severity of cardiovascular disorders [3]. The underlying pathogenesis of this relationship may be various and includes hemodynamic changes, sympathetic hyperactivity, endothelial dysfunction, systemic inflammation, and oxidative stress [4]. Therefore, OSA has been shown to be an important risk factor for cardiovascular diseases, including coronary artery disease, myocardial infarction, heart failure (HF), arterial hypertension, cardiac arrhythmias, pulmonary hypertension, and stroke [5].

### 1.1. OSA Impact on Cardiovascular System

Previous studies have shown that OSA causes the remodeling of the left ventricle (LV) and left atrium (LA). Increased LV pressure and diastolic dysfunction cause LA enlargement and structural remodeling [6]. Recently, the potential impact of OSA on the development of LV diastolic dysfunction (including right ventricle (RV) diastolic function) has been increasingly emphasized [7]. However, the data on this relationship are still conflicting, as several coexisting OSA disorders may also affect the diastolic properties of the LV myocardium (e.g., arterial hypertension, diabetes mellitus) [8,9]. OSA affects LV systolic function as a result of reduced LV preload and increased LV afterload; however, in the early stages of the disease, LV dysfunction is subclinical. If untreated, OSA leads to the progression of LV hypertrophy and HF decompensation associated with increased morbidity and mortality [6].

OSA is not only complicated by left heart disease but is usually associated with pulmonary hypertension [10]. According to the World Health Organization (WHO), PH can be classified into five clinical groups with specific features. Group 2 is the most common reason for PH and is due to left heart disease: systolic and diastolic dysfunction, as well as valvular disease [10,11]. The mechanisms of PH in OSA consist of several factors. OSA causes PH via the mechanism of hypoxia by activating vasoactive factors and hydrostatic mechanisms due to an increase in left atrial (LA) pressure leading to pulmonary venous hypertension. Both pathways finally cause vascular remodeling, pulmonary arterial hypertension, and right heart dysfunction [12]. PH due to hypoxemia belongs to Group 3 according to WHO PH classification, so OSA is especially associated with this group. Increased pulmonary artery pressure leads to RV hypertrophy, while elevated venous return causes volume overload related to RV remodeling, both of which lead to RV dysfunction [6].

### 1.2. OSA and Blood Serum Biomarkers

OSA is diagnosed during overnight polysomnography and treated with continuous positive airway pressure (CPAP). Even when OSA is diagnosed, issues such as poor patient adherence can prevent adequate treatment with CPAP overnight. Because OSA is associated with cardiovascular diseases, echocardiographic evaluation is the preferred diagnostic test. Considering that OSA is associated with cardiovascular dysfunction, an ideal blood serum biomarker should allow for the timely recognition of increased cardiovascular risk and help in the early diagnosis and treatment of cardiovascular complications. More and more new results are coming on the utility of serum biomarkers for assessing disease severity, prognosis, and response to treatment. However, blood serum biomarkers, which could evaluate CPAP treatment efficacy for the cardiovascular system, have not yet been widely investigated. Various studies have searched for biomarkers for the screening of OSA [13,14,15], but it is still not clear which biomarkers could assess CPAP treatment efficacy and its impact on the cardiovascular system. In our clinical trial, we selected and investigated these biomarkers, which are the most predictive and associated with cardiovascular disorders in patients with OSA.

#### 1.2.1. Galectin-3

Galectin-3 is a biomarker of fibrosis and inflammation and has been implicated in the development and progression of HF, and it may predict increased morbidity and mortality [16]. However, galectin-3 is not a specific cardiac biomarker, and its concentration depends on renal function [17]. Singh et al. investigated the association between galectin-3 levels and OSA severity and its impact on myocardial fibrosis, which leads to increased cardiovascular diseases [18]. Pusuroglu et al. concluded that this biomarker is associated with coronary atherosclerosis and OSA severity [19], though we did not find any data on changes caused by this biomarker during CPAP treatment.

#### 1.2.2. sST2

Interleukin-33 is a functional ligand of ST2, involved in reducing fibrosis and hypertrophy in mechanically stressed tissues. Recent studies have demonstrated soluble ST2 to be a strong independent predictor of cardiovascular outcomes in both chronic and acute HF [20]. In contrast to NT-proBNP, sST2 is not related to age, BMI, renal function, or causes of HF [17]. There are only a few studies on the association between the sST2 biomarker and OSA. Kopeva et al. investigated sST2′s prognostic value in HF patients with OSA [21]. sST2 level changes were investigated in Andreieva’s scientific work, which showed that sST2 significantly decreased only in the CPAP group after 6 months of treatment [22]. More precise studies on the effect of CPAP treatment on this biomarker and its relationship with cardiovascular diseases are still needed.

#### 1.2.3. Endothelin-1

ET-1 is one of those more widely discussed biomarkers in OSA compared with galectin-3 and sST2. Harańczyk et al. investigated OSA’s influence on endothelial dysfunction, and one of their conclusions was that long-term CPAP treatment does not have any impact on changes in biomarker levels [23]. Lin et al. performed a meta-analysis showing that CPAP treatments in OSA patients were significantly related to a decrease in ET-1 levels [24]. However, further long-term studies with a bigger population are still needed to clarify these findings, as the effectiveness of CPAP on ET-1 levels has had contradictory results.

The early detection of left or right ventricle dysfunction with echocardiographic techniques may be useful in early applications of CPAP and preventing the progression of LV hypertrophy, diastolic and systolic dysfunction, and pulmonary hypertension, as the later development of HF leads to cardiovascular death. Two-dimensional (2D) echocardiography is the gold standard for the evaluation of the cardiac remodeling associated with OSA. Conventional echocardiography is not sensitive enough to detect early cardiac dysfunction in OSA patients; thus, 2D speckle-tracking (ST) echocardiography is an informative and promising method of detecting subclinical changes in LV or LA function. In addition, it is very important to evaluate CPAP treatment effectiveness by researching parameters reflecting LV remodeling. It likely that the identification of blood serum biomarkers could be the most accurate reflection of these changes.

## 2. Materials and Methods

### 2.1. Subjects

This was a longitudinal study that assessed echocardiographic parameters and particular blood serum biomarkers in OSA patients before and after 3 months of CPAP therapy. First part of this study (the same clinical trial subjects) was the evaluation of the quality of life in OSA patients during CPAP treatment [25].

From January 2020 to June 2021, 34 participants with moderate-to-severe OSA were included in the trial. After receiving CPAP treatment for 3 months, 17 individuals underwent the final investigation. Thirteen patients had their 2D ST echocardiographic parameter pre- and post-CPAP treatment evaluated. Patients who dropped out were either lost in the follow-up or denied CPAP therapy.

The study was authorized by the Kaunas Regional Biomedical Research Ethics Committee (no. BE-2-23, 19 May 2020, Kaunas, Lithuania).

Age 18 to 65, a diagnosis of moderate to severe OSA, a body mass index equal or more than 30 kg/m^2^, and a completed informed consent form were the inclusion criteria.

Subjects under the age of 18 and adults over the age of 65 were excluded, as well as people with significant mental and/or internal organ disorders that might have an impact on the study’s protocol but who did not have clinically significant ischemic heart disease, severe valvular heart disease (grade 3), or uncontrolled arterial hypertension (AH). The exclusion criteria also included the absence of a signed informed consent form and the investigator’s discretion.

All individuals received thorough clinical investigations, which included thorough physical examinations and the documentation of symptoms, medical and surgical histories, and comorbidities. At the Hospital of Lithuanian University of Health Sciences (LUHS) Kaunas Clinics Outpatient Clinic, patients were examined and had otorhinolaryngologist consultations to rule out any nasal pathology (e.g., nasal polyposis, deviated nasal septum, insufficiency of the nasal valve).

People who were thought to have OSA were referred to the LUHS Kaunas Clinics Pulmonology Department’s Sleep Laboratory for an overnight diagnostic polysomnography (PSG). Using the Alice 6 LDx diagnostic sleep system, this investigation was conducted (Philips Respironics, Murrysville, PA, USA). Apnea was defined as the absence of airflow for longer than 10 s, and hypopnea as a decrease in airflow for at least 10 s that was also accompanied by a 3% drop in SpO2 or arousal. During the study, the apnea-hypopnea index (AHI) was measured for each hour of sleep. According to the AHI, OSA patients were divided into three categories: mild OSA (AHI ≥ 5 but <15), moderate OSA (AHI ≥ 15 but <30), and severe OSA (AHI ≥ 30) [5,6].

Each patient spent an additional night in the sleep laboratory following the diagnosis of OSA for CPAP titration, which determines the ideal pressure at which the CPAP equipment eliminates aberrant breathing occurrences. To take part in the clinical investigation, patients with moderate or severe OSA were invited.

The control group consisted of 13 subjects. Control group subjects underwent all the same diagnostic steps as the OSA group. The inclusion criteria were: age 18–65 years old, diagnosis of mild OSA (AHI ≥ 5 but <15) or no OSA diagnosis during overnight diagnostic PSG (AHI < 5), body mass index (BMI) ≥ 30 kg/m^2^, and signed informed consent form.

### 2.2. Sample Collection and Biomarker Testing

As mentioned before, three biomarkers were tested: galectin-3, sST2, and ET-1. Peripheral blood samples were collected in BD Vacutainer^®^ (BD Bioscience, San Jose, CA, USA) serum tubes at the time of study enrolment before the initiation of CPAP therapy and after 3 months of CPAP treatment (blood samples were taken only for OSA group patients). After collecting the blood, samples were allowed to clot for 1 h at room temperature and, after that time, centrifuged at 1600× *g* for 10 min. Following centrifugation, serum was separated from the blood and transferred into clean 1 mL cryotubes using a Pasteur pipette. Whole blood samples were stored at −80 °C in a refrigerator.

The serum concentrations (ng/mL) were analyzed using a commercially available enzyme-linked immunosorbent assay (ELISA Kit, Elabscience^®^, Houston, TX, USA) according to the manufacturer’s instructions.

### 2.3. Echocardiography

2D transthoracic echocardiography, including ST, was performed for all patients using an EPIQ 7 (Phillips Ultrasound Inc., Bothell, WA, USA) ultrasound machine and an M3S cardiac 4.0 MHz transducer. Two experienced cardiologists conducted all conventional measurements and 2D transthoracic echocardiography studies.

Conventional measurements were taken in accordance with the American Association of Echocardiography’s recommendations. [26]. LV end-diastolic diameter (EDD) and LA diameter were evaluated from the parasternal long-axis view, while basal right ventricular diameter (RVD1) and right atrial (RA) diameter were measured from the RV-focused four-chamber view at the end-diastole and end-systole, respectively. LA maximal volume was obtained with the area-length method. Myocardial mass (MM) was determined using the Devereux formula. According to the newest guidelines for obese patients, the LA volume and LV myocardial mass were indexed by height (normal LA volume ≤ 18.5 mL/m^2^ for men and ≤16.5 mL/m^2^ for women; normal LV myocardial mass ≤ 50 g/m^2.7^ for men and ≤47 g/m^2.7^ for women) [27].

At the apical four- and two-chamber views, the LV ejection fraction (EF) was estimated using Simpson’s biplane approach. By evaluating the mitral diastolic flow velocity ratio (E/A), the E/e ratio, the volume index of the LA, and the tricuspid regurgitation velocity, LV diastolic function was assessed in accordance with established recommendations [28].

The peak systolic velocity of the tricuspid annulus (RV S′), as measured by Tissue Doppler imaging, and the RV fractional area change (FAC), as determined by tracing the RV endocardial boundaries at the end-diastole and end-systole, were used to evaluate the RV longitudinal function [26].

The longitudinal strains of LV, LA, and RV were assessed with ST echocardiography using the TomTec GmbH (Munich, Germany) imaging system according to existing guidelines [29]. The LV longitudinal strain was evaluated from 4-, 3-, and 2-chamber apical views using the auto LV strain automatic analysis function. LA deformation parameters were obtained in four-chamber apical views using the auto LA strain automatic analysis function. Only LA reservoir strain (positive value) was analyzed. RV longitudinal strain was obtained from the RV- focused four-chamber view using the auto RV strain function. Mean pulmonary artery (PA) pressure was calculated using the formula mPAP = 80 − (0.5 × AT_RVOT_); the normal mean PA pressure was defined as <25 mmHg [30].

### 2.4. Interventions and Follow-Up

Study subjects were asked to return for a follow-up visit 3 months after treatment. CPAP treatment compliance was checked by downloading data from the CPAP device. We defined compliance with treatment as using the CPAP device for more than 4 h per night for more than 70 percent of nights and an AHI < 5.

After 3 months of CPAP treatment, a 2D ST echocardiography was performed and blood samples were taken.

For the control group, a 2D ST echocardiography was performed just once as a baseline (as no treatment with CPAP was initiated).

### 2.5. Statistical Analysis

We analyzed both groups of patients: control group (no OSA was diagnosed or it was mild) and those with continuous CPAP treatment (OSA group; values at a baseline and 3 months after CPAP treatment initiation). Quantitative variables were described using the mean and standard deviation or median at the 25–75th percentiles. To describe the distribution and changes in all quantitative variables, the nonparametric Mann–Whitney U-test for two dependent samples was used.

Receiver operating characteristic (ROC) curve analysis was conducted to describe the predictive value of strain parameters; the correspondent area under the curve with its 95% confidence interval was calculated.

IBM SPSS Statistics for Windows, version 27.0 (IBM Corp., Armonk, NY, USA), was used for statistical analysis. Statistical significance was considered to be *p* < 0.05.

## 3. Results

### 3.1. Patients Characteristics

A total of 34 patients diagnosed with OSA were enrolled in the study. The control group consisted of 13 subjects. The incidence of OSA was higher for men compared to women (odds ratio (OR) 4.97; 95% confidence intervals CI 1.17–21.10; *p* = 0.03). OSA group subjects were older compared to the control group. BMI did not differ significantly between either group. In total, 26 OSA group patients and 10 control group subjects had various chronic diseases, but the most common were AH and diabetes mellitus (Table 1).

Based on the receiver operating characteristic (ROC) test (Figure 1), we obtained groups with significant age differences. If the respondent was >44 years old, OR for OSA was higher at 5.44 (95% CI 1.34–22.13) (Table 2).

### 3.2. Echocardiographic Parameters

LV end-diastolic diameter and volume, as well as LV EF, did not differ between the groups, but increased LV end-systolic volume and reduced LV global longitudinal strain (GLS) were found in OSA group patients (*p* = 0.015 and *p* = 0.035 respectively). Indexed by height, a higher LV Mmi in OSA patients (*p* = 0.007) and a higher prevalence of LV diastolic dysfunction (*p* = 0.023) were found in this group of patients. Though the LA volume did not differ between groups, the OSA group patients had significantly lower LA reservoir strain (*p* < 0.001).

Conventional RV longitudinal and global function parameters (S′, fractional area change (FAC)) did not differ between the groups; however, the RV GLS was reduced in OSA patients (*p* = 0.026). OSA patients had a significantly higher RA diameter and mean PAP (*p* < 0.05) (Table 3).

Table 4 summarizes changes in biventricular geometry and function, as well as the remodeling of atriums in OSA group patients before and 3 months after CPAP treatment initiation.

The LV diameter, Mmi, and LV ejection fraction did not change after treatment. The LV global longitudinal strain improved, as well as the LV diastolic function parameters. We found that LV diastolic dysfunction was more prevalent in the OSA group. CPAP treatment improved LV diastolic function after 3 months.

We found no LA volume changes after treatment, but LA reservoir function significantly improved 3 months after treatment with CPAP (*p* = 0.008). Right ventricle geometry, function, and deformation parameters did not change.

### 3.3. Blood Serum Biomarkers

The galectin-3 and sST2 concentrations significantly decreased after 3 months of CPAP treatment; however, we did not find significant changes in ET-1 concentration (Table 5).

## 4. Discussion

In this study, we analyzed the effect of short-term CPAP treatment on LV and LA geometry and function using 2D ST echocardiography and used blood serum biomarkers (galectin-3, sST2, and endothelin-1) as an element and sign of CPAP treatment efficacy.

Our results show that OSA is related to increased LV end-systolic volume and LV MMi (as indexed by height) and is more prevalent in LV diastolic dysfunction. Although the conventional echocardiographic parameters of LV and RV function did not differ, 2D ST echocardiography revealed subclinical dysfunction in both ventricles and LA, as the LV GLS, RV GLS, and LA reservoir strain were significantly lower in OSA patients compared to the controls. CPAP treatment had a positive effect on LV longitudinal and LA reservoir function (LV GLS and LA reservoir strain significantly improved) and was related to a significant reduction in serum biomarker (galectin-3, sST2) concentration.

Various studies have searched for possible useful biomarkers that can be used for OSA screening; however, there are only a few that could be used to evaluate the efficacy of CPAP treatments for the cardiovascular system in OSA patients. CPAP is an efficient treatment for OSA, but adherence remains challenging. Response to treatment is quite complicated, as patients usually feel only subjective component changes, such as those relating to daytime symptoms. Thus, it is very important to show improvement in a patient via objective data—for example, serum biomarker reduction and cardiac function improvement, as was investigated in our study. Furthermore, different clinical trials have had different perspectives on the evaluation of the cardiovascular system, as the same echocardiographic parameters were not used.

It is very difficult to describe the necessary criteria for perfect OSA biomarkers; Montesi et al. [13] looked for them in general, covering OSA pathogenesis in particular, and concluded that no ideal biomarker exists at this time.

Galectin-3 is a biomarker of fibrosis and inflammation and is involved in the development and progression of HF, and it may predict increased morbidity and mortality. Several meta-analyses have shown that increased galectin-3 expression levels are associated with mortality and acute and chronic HF [31]. An interesting study was published by Singh et al. The main finding of their research was that moderate–severe OSA is associated with increased galectin-3 concentrations, and these elevated levels are more likely in women than in men [18]. In our study, gender was not segregated, but an increase in galectin-3 concentration was observed in the overall group. In 2018 in the European Respiratory Journal, Andreieva published a scientific thesis in which the effect of short-term CPAP treatments on galectin-3 concentrations were investigated [32]. The researcher did not find any significant galectin-3 level changes during a 3-month CPAP treatment period, but BMI was a meaningful factor in galectin-3 concentrations, which is inconsistent with our study. In our study, BMI remained the same, so we believe and can conclude that treatment with CPAP was the main influence on galectin-3 level changes. Ansari et al. conducted a clinical trial investigating the association between galectin-3 level and echocardiographic parameters. The study aimed to investigate the importance of galectin-3, evaluating echocardiographic parameters for patients with HF but preserved EF [33]. This study provides useful information, proving the association between LV structure and function and galectin-3 levels, essentially reflecting the hypothesized link between cardiac fibrosis, hypertrophy, and evolving HF. The study data of Andrejic et al. confirms that galectin-3 levels are higher when associated with left ventricular remodeling [34]. Galectin-3 does not provide differentiation between HF with preserved or reduced EF, but it reveals the severity of diastolic dysfunction and ventricular stiffness in patients with preserved LV EF [17]. One more revealing scientific thesis was published in the Journal of Hypertension by Ionin et al. They found that increased galectin-3 concentrations are higher to patients with OSA and atrial fibrillation. It is supposed that the profibrogenic activity of galectin-3 may induce the risk of atrial fibrillation [35]. There are only a few clinical trials that have investigated the association between galectin-3 concentration changes and echocardiographic parameters in OSA patients treated with CPAP. Cicco et al. investigated galectin-3 in relation to heart damage in patients with OSA and its role in inflammation [36]. They found that galectin-3 concentrations were higher in the severe OSA group, as was noted in our study. What is more, galectin-3 is related to LV echocardiographic parameters and is associated with cardiac remodeling. Our study did not investigate the correlation between blood serum biomarkers and echocardiographic parameters due to the small sample of subjects, so this is a topic for further clinical trials.

The prognostic value of sST2 in HF patients with OSA was discussed in the clinical study by Kopeva et al. The main finding was that sST2 might be used as a diagnostic biomarker for the prognosis of adverse cardiac events in HF patients with preserved EF, as higher sST2 concentration levels are associated with a higher frequency of adverse cardiac events in follow-up periods [21]. Daniels et al. investigated sST2 levels association with cardiac function and structure. The main finding was that the presence of right ventricular hypokinesis independently associated with sST2 levels [37]. Farcas et al. published a study analyzing the relationship between sST2 and diastolic dysfunction [38]. The main result of their research was that sST2 levels correlate with the parameters of LV remodeling and diastolic dysfunction. In general, according to other researchers and our findings—i.e., that sST2 concentration levels significantly change after CPAP treatment initiation—sST2 might be used as a biomarker for cardiac remodeling.

ET-1 is a mediator that participates in endothelial dysfunction progress caused by the irreversible process of vascular remodeling, resulting in an effect on systemic and pulmonary circulation. [39]. As a result, ET-1 has a major role in the development of pulmonary hypertension. This hypothesis was proved in the study by Carratu et al. The main finding was that ET-1 concentration levels significantly correlate with systolic PAPs in obese patients, both with and without OSA [40]. The association between elevated ET-1 concentrations and PH development was noticed in the Jackson heart study [41], but the clinical trial subjects were not OSA patients. Long-term CPAP treatment effects, ET-1 changes, and echocardiographic correlations were investigated and discussed in Harańczyk’s study [23]. It proved other researchers’ hypothesis that ET-1 is associated with the development of PH. Nevertheless, there are only a few studies that have investigated ET-1′s relationship with LV function.

It has been observed that OSA group patients have a more prevalent AH ratio. Periodic hypoxemia and hypercapnia due to apnea–hypopnea episodes cause sympathetic nerve activation and elevate serum catecholamine, leading to an increase in heartrate and blood pressure. Over time, these hemodynamic changes lead to LV hypertrophy, diastolic dysfunction, and HF [42,43]. A previous study revealed that OSA might change LV geometry independently of AH [44]. In our study, the OSA group had a significantly higher AH ratio; however, it was well controlled. Thus, we might hypothesize that LV hypertrophy and remodeling are mainly due to OSA-related changes.

Detailed echocardiographic parameters in OSA patients, but without treatment options, were overviewed in a review by Sascau et al. [45]. The main findings were that both moderate and severe OSA are associated with decreased ventricular function and increased atrial volume, leading to a high incidence of chronic HF and atrial fibrillation in patients. The first meta-analysis from Yu Lei et al. showed that OSA is associated with LV and LA remodeling (increased LV end-diastolic diameter, LV end-systolic diameter, LV myocardial mass, LA diameter, and LA volume index) [6].

The treatment of OSA with CPAP reduces hypoxia episodes; improves hemodynamic parameters such as arterial stiffness, blood pressure, and pulmonary vasculature resistance; and normalizes hormonal (sympathetic hyperactivity) as well as metabolic alterations [46]. The effects of CPAP on cardiac remodeling were widely discussed in a study by Colish et al.: After 3 months of CPAP initiation, improvements were seen in RA and RV diameter, LA diastolic function (LA volume reduction), LV filling pressures assessed by echocardiography, and LV mass by CMR. The main conclusion of their research was that CPAP treatment has a positive effect on cardiovascular remodeling [47].

Meta-analyses by Sun et al. reviewed the impact of CPAP on LV EF in patients with OSA [48]. A significant improvement in LV EF was observed after CPAP treatment, especially in patients with HF. However, some previous studies have not shown better cardiovascular outcomes, which might be due to a late start of therapy after irreversible myocardial damage was diagnosed [49,50]. Egea et al. published a randomized clinical trial that investigated cardiac function after CPAP therapy for 3 months [42]. The subjects of this trial were patients with an LV EF < 45% (patients were divided into groups according to LV EF). The main finding was that, after 3 months of treatment with CPAP, LV EF significantly improved for patients with LV EF > 30%. Other echocardiographic parameters did not change. In our study, all patients had preserved LV EF, and after treatment with CPAP, LV EF did not change significantly. Thus, according to this study and based on our research data, we might hypothesize that CPAP is more effective with subclinical LV dysfunction.

The results of a recent meta-analysis showed that CPAP treatment improves the biventricular function of OSA patients, and ST echocardiography should be a routine diagnostic test for these patients rather than conventional echocardiography due to its limitations [51]. According to the authors, LV global longitudinal strain (GLS) and RV GLS are more sensitive than conventional echocardiographic parameters (LV EF, tricuspid annular plane systolic excursion (TAPSE)) in detecting biventricular dysfunction in OSA patients. Moreover, these parameters are reliable measurements for CPAP treatment efficacy.

Interesting results were published in a study by D’Andrea et al. [52]. The authors assessed the acute and chronic effects of noninvasive ventilation (NIV) on left and right myocardial functions using ST echocardiography on patients with OSA. In the acute treatment phase, LV EF, diastolic parameters, and LV regional peak myocardial strain improved; meanwhile, RV GLS, RV regional peak myocardial strain, and RA lateral wall strain significantly decreased, and RA diameter and area and pulmonary artery systolic pressure (PASP) increased. At a follow-up after 6 months of treatment, PASP, RV, and RA deformation parameters improved, showing reversible RV and RA dysfunction during NIV. These findings might be explained by RV diastolic function impairment, increased RV afterload, and the reduced return of venous blood during NIV. In our study, a 3-month CPAP treatment was related to LV GLS and LA reservoir function improvement; conversely, the RA diameter and the mean PAP significantly increased, and RV GLS did not change, possibly due to the too short treatment period for the right heart to adapt to NIV, as well as subacute alterations in the pulmonary vasculature and RA and RV performance.

Previous studies have revealed that, in patients with OSA, repetitive hypoxia episodes during sleep cause myocardial ischemia, increased afterload, and subclinical LV dysfunction, especially longitudinal LV deformation impairment [53]. Kim et al. showed that CPAP treatment for 3 months significantly improves LV GLS in OSA patients [54], thus suggesting the possibility of preventing irreversible LV dysfunction with early CPAP treatment. The independent predictor of LV longitudinal dysfunction was hypoxia rather than AHI, gender, or BMI.

OSA is associated with LA remodeling and dysfunction, occurring even before LV hypertrophy and LV diastolic dysfunction. Wan et al. showed that the LA reservoir and conduit function reduce in OSA patients irrespective of LV hypertrophy presence; meanwhile, active pump function increased [55]. According to previous studies, CPAP treatment increases LA GLS, showing the reversible remodeling of LA in the early stages of OSA [56]. The results of a study by Vural et al. revealed that a 24-week treatment with CPAP is associated with significantly decreased E/e′ ratios and LA volumes and significantly increased LA reservoirs, conduit strain, and strain rates [57]. In concordance with other studies, our research showed that LA diameter was significantly higher for OSA patients, while LA volume did not differ even when indexed by height. After 3 months of CPAP treatment, neither LA diameter nor LA volume changed, but LA reservoir strain significantly improved. ESC guidelines on the management of arterial hypertension recommend height-based indexing to define LV hypertrophy (LV mass/height in m^2.7^), as height-based indexing preserves the effect of obesity on LV mass [44]. In our study, there was no difference in LV myocardial mass when indexed by body surface area, while significant differences were obtained when it was indexed by height.

According to our clinical study results, subclinical dysfunction in RV was found. PH is diagnosed in approximately 40 percent of OSA patients [11]. Generally, this PH is mild or moderate in severity, but it can determine RV hypertrophy and dysfunction [58]. Previous studies have shown that conventional echocardiographic parameters (TAPSE, S′, RV FAC) are insufficiently sensitive in detecting subclinical RV dysfunction; therefore, the determination of longitudinal strain in the RV becomes extremely important [58]. During CPAP treatment, longitudinal strain in the RV is mostly found in the apical segments, so the total longitudinal strain of the RV may not change significantly. In our study, PH was found in 56 percent of patients (mean pulmonary artery pressure ≥ 25 mmHg). The conventional echocardiographic parameters of OSA patients did not differ from the control group, but we found a significantly lower RV longitudinal strain with ST echocardiography. After 3 months of CPAP treatment, it did not differ significantly, possibly because we did not evaluate regional RV longitudinal strain.

## 5. Limitations of the Study

This clinical trial has some limitations that should be highlighted. The main limitation of this study is the relatively small patient sample. Both groups of our study patients (those who did not start or who discontinued CPAP treatment and those who used the CPAP device and had good compliance) were similar to each other. We believe that patients enrolled in the study could be representative of all patients with moderate or severe OSA. One more limitation is that blood serum biomarkers were not tested for in the control group (patients with mild OSA or without an OSA diagnosis). Thus, it was not possible to compare data between healthy subjects and make relevant conclusions.

## 6. Conclusions

OSA is associated with the left heart remodeling process—increased LV myocardial mass index, LV diastolic dysfunction, reduced LV and RV longitudinal strain, and reduced LA reservoir function. A short-term, 3-month CPAP treatment does not change the geometry of left heart chambers, but it does improve LV global longitudinal strain and LA reservoir function and positively affects blood serum biomarkers. This new indexing system for LV myocardial mass via height helps to identify myocardial structural changes in obese patients with OSA. Further research is needed to investigate galectin-3, sST2, and ET-1 correlations with echocardiographic parameters in treatment with CPAP.

## Figures and Tables

**Figure 1 medicina-58-01511-f001:**
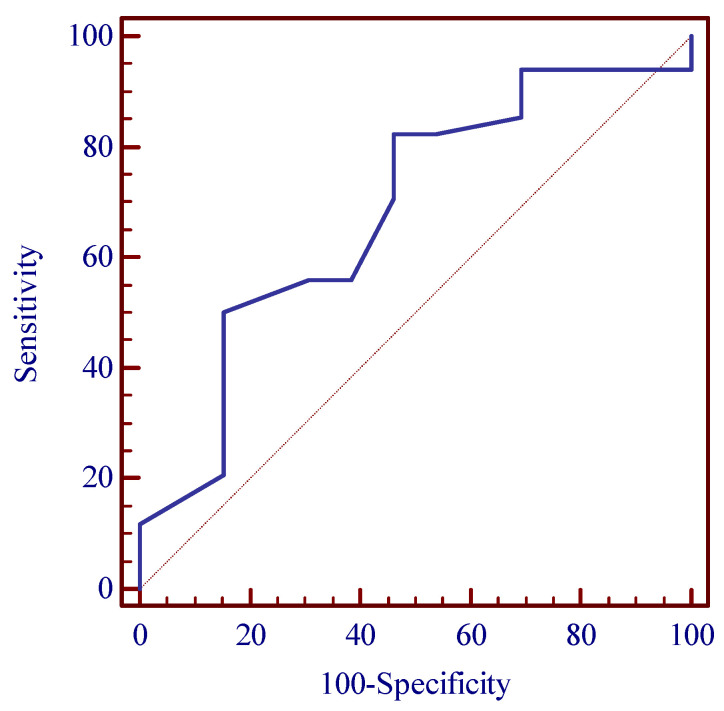
Respondent ROC curve for predicting the threshold age.

**Table 1 medicina-58-01511-t001:** Characteristics of OSA and control group subjects.

		OSA Group (n = 34)	Control Group (n = 13)	*p*-Value *
		Mean (Standard Deviation)		
Sex	Men (%)	29 (85.3)	7 (53.8)	0.023 *
	Women (%)	5 (14.7)	6 (46.2)	0.76
Age, years		52.15 (8.89)	45.46 (9.47)	0.028 *
BMI, kg/m^2^		37.94 (5.31)	35.33 (5.32)	0.108
Chronic diseases	Arterial hypertension, n (%)	26 (76.5)	5 (38.5)	0.014 *
	Diabetes mellitus, n (%)	1 (8.3)	8 (23.5)	0.254
	Asthma, n (%)	1 (7.7)	1 (2.9)	0.481
Smoking status, n (%)		21 (61.8)	6 (46.2)	0.33
AHI, per h		66.94 (25.33)	7.02 (4.45)	0.00 *

* *p*-Value < 0.05, according to nonparametric Wilcoxon test. OSA: obstructive sleep apnea, AHI: apnea–hypopnea index, BMI: body mass index.

**Table 2 medicina-58-01511-t002:** ROC test for the predicted values of respondent ages and the distribution of its characteristics.

Parameter/Threshold Value	Area under the ROC Curve (AUC) (%)	Sensitivity/Specificity (%)	Control Group/OSA Group N (%)	*p*-Value *	OR for OSA Incidence [95% CI]
Age >44 years	68.0	82.4 53.8	6 (46.2) 28 (82.4)	0.013 *	5.44 [1.34–22.13]

* *p*-Value < 0.05, according to nonparametric Wilcoxon test.

**Table 3 medicina-58-01511-t003:** Echocardiographic parameters of control and OSA group patients.

		Control Group (n = 13)	OSA Group (n = 34)	*p*-Value *
LV EDD (mm)		47.62 ± 3.97	50.08 ± 4.69	0.165
LV EDV (mL)		102.55 ± 26.05	142.25 ± 26.84	0.026 *
LV ESV (mL)		44.08 ± 12.72	67.06 (14.77)	0.015 *
LV Mmi (g/m^2^)		79.56 ± 2.92	88.98 ± 13.51	0.084
LV Mmi by height (g/m^2.7^)		38.96 ± 5.90	46.11 ± 8.23	0.007 *
IVS (mm)		10.6 ± 1.09	11.42 ± 1.32	0.006 *
LV EF (%)		56.45 ± 2.35	54.15 ± 3.59	0.055
LV GLS (%)		−23.20 ± 2.44	−16.56 ± 3.31	0.035 *
LA diameter (mm)		36.23 ± 3.08	43.58 ± 8.19	<0.001 *
LA volume (mL)		64.35 ± 3.09	70.97 ± 13.80	0.191
LA dilatation (volume index by BSA, >34 mL/m^2^)		1 (7.7)	5 (14.70)	0.324
LA dilatation (volume index by height >18.5 mL/m^2^ men, >16.5 mL/m^2^ women)		9 (69.2)	19 (55.88)	0.289
LV diastolic dysfunction	Normal	7 (53.8)	4 (15.4)	0.781
	Grade 1	6 (46.2)	22 (80.8)	0.023 *
	Grade 2	0 (0)	1 (3.8)	0.923
	Grade 3	0 (0)	0 (0)	0
LA reservoir strain (%)		31.01 ± 1.56	26.56 ± 1.38	<0.001 *
RV diameter (mm)		35.46 ± 4.23	36.55 ± 4.57	0.419
RA diameter (mm)		37.62 ± 4.48	40.58 ± 3.4	0.043 *
RV S′ (cm/s)		13.18 ± 2.27	14.06 ± 6.16	0.941
RV FAC (%)		43.72 ± 5.92	41.51 ± 4.66	0.415
RV GLS (%)		−25.03 ± 3.21	−19.65 ± 3.75	0.026 *
Mean PAP (mmHg)		22.19 ± 7.60	28.59 ± 7.08	0.037 *

Data presented as mean ± SD (standard deviation), N (%). * *p*-Value < 0.05, according to nonparametric Wilcoxon test. OSA: obstructive sleep apnea, LV: left ventricle, EF: ejection fraction, GLS: global longitudinal strain, EDD: end-diastolic diameter, ESV: end-systolic volume, EDV: end-diastolic volume, MM: myocardial mass, LA: left atrial, RV: right ventricle, RA: right atrium, FAC: fractional area change, PAP: pulmonary artery pressure, IVS: interventricular septum.

**Table 4 medicina-58-01511-t004:** Changes in echocardiographic parameters in OSA group patients after treatment with CPAP.

		OSA Group (n = 13)		
		Before Treatment	3 Months after CPAP Treatment	*p*-Value *
		Mean (Standard Deviation)		
LV EDD (mm)		50.08 (4.69)	49.75 (3.57)	0.504
LV EDV		142.25 (26.84)	127.08 (33.8)	0.022 *
LV ESV (mL)		67.06 (14.77)	53.74 (17.8)	0.037 *
LV EF (%)		54.15 (3.59)	54.83 (3.27)	0.473
LV Mmi (g/m^2^)		88.98 (13.51)	80.06 (20.79)	0.963
LV Mmi by height (g/m^2.7^)		46.11 (8.23)	43.63 (6.29)	0.480
LV GLS (%)		−16.28 (3.82)	−18.82 (3.04)	0.005 *
LA reservoir strain (%)		25.82 (7.6)	32.45 (5.64)	0.008 *
LA volume (mL)		64.35 (3.09)	67.56 (11.24)	0.328
LA diameter (mm)		43.58 ± 8.19	43.79 ± 2.27	0.874
LV diastolic dysfunction	Normal	4 (15.4)	6 (46.15)	0.657
	Grade 1	22 (80.8)	6 (46.15)	0.024 *
	Grade 2	1 (3.8)	1 (7.7)	0.973
	Grade 3	0 (0)	0 (0)	0
RV diameter (mm)		36.55 (4.57)	39.17 (3.83)	0.411
RA diameter (mm)		40.58 (3.4)	41.92 (3.75)	0.623
RV S′ (cm/s)		14.06 (6.16)	13.62 (3.72)	0.624
RV FAC (%)		41.51 (4.66)	43.49 (7.68)	0.104
RV GLS (%)		−19.65 (3.75)	−21.15 (4.57)	0.608
Mean PAP (mmHg)		28.59 (7.08)	29.62 (8.19)	0.075

* *p*-Value < 0.05, according to nonparametric Wilcoxon test. OSA: obstructive sleep apnea, CPAP: continuous positive airway pressure, EF: ejection fraction, GLS: global longitudinal strain, RS: reservoir strain; EDD: end-diastolic diameter, MM: myocardial mass, LA: left atrial, RV: right ventricle, RA: right atrium, FAC: fractional area change, PAP: pulmonary artery pressure, E: early diastolic transmitral velocity, E′: early diastolic mitral annular velocity, S′: myocardial systolic excursion velocity, IVS: interventricular septum, E/A: the ratio of the early € to late (A) ventricular filling velocities, LV LS: left ventricular longitudinal strain.

**Table 5 medicina-58-01511-t005:** Changes in blood serum biomarker concentrations after treatment.

	Before Treatment	3 Months after CPAP Treatment	*p*-Value *
	Mean (Standard Deviation)		
Galectin-3 (ng/mL)	17.52 ± 1.19	11.64 ± 0,97	0.001 *
sST2 (ng/mL)	0.56 ± 0.47	0.41 ± 0.56	0.047 *
ET-1 (ng/mL)	3.82 ± 2.27	3.56 ± 2.27	0.28

* *p*-Value < 0.05, according to nonparametric Wilcoxon test. CPAP: continuous positive airway pressure, ET-1: endothelin-1.

## Data Availability

Not applicable.

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
