# Peer review of "Changes in Left Heart Geometry, Function, and Blood Serum Biomarkers in Patients with Obstructive Sleep Apnea after Treatment with Continuous Positive Airway Pressure"

_medicina, 2022, doi:10.3390/medicina58111511_

Round 1

Reviewer 1 Report

The authors aim to evaluate left heart geometry, function, deformation parameters, and blood serum biomarker (galectin-3, sST2, endothelin-1) levels in patients with OSA and assess changes after short-term CPAP treatment. The study is very well-written.  A few suggestions are listed below:

1. The level of sleep study that was carried out

2. Why was patients' overall QOL post CPAP not included?

3. Include a statement on CPAP compliance and the effect of the study's objective blood serum biomarker and left heart geometry

4. Highlight possibility of complications with CPAP usage

Author Response

Dear Reviewer, thank you very much for your detailed analysis of our manuscript and for the dedicated time. We really appreciate your help. We would like to answer your questions and to make clarification of all your comments, suggestions and observations.

  1. The level of sleep study that was carried out

All clinical trial subjects underwent full whole night polysomnography, in order to diagnose obstructive sleep apnea and to evaluate the severity of it.

  1. Why was patients' overall QOL post CPAP not included?

The Quality of life for OSA patients was evaluated and described earlier. The main purpose of our study was to evaluate the cardiovascular changes during CPAP treatment. The QOL results can be found in our published manuscript “Short-Term Continuous Positive Air Pressure Treatment: Effects on Quality of Life and Sleep in Patients with Obstructive Sleep Apnea” (DOI: 10.3390/medicina58030350).

  1. Include a statement on CPAP compliance and the effect of the study's objective blood serum biomarker and left heart geometry

In our pilot study we did not evaluate CPAP compliance effect to blood serum markers and echocardiographic parameters.

  1. Highlight possibility of complications with CPAP usage

Common problems with CPAP include a leaky mask, trouble falling asleep, a stuffy nose and a dry mouth. During our study, we did not observe any of these CPAP usage problems or complications.

Author Response

Dear Reviewer, thank you very much for your detailed analysis of our manuscript and for the dedicated time. We really appreciate your help. We would like to answer your questions and to make clarification of all your comments, suggestions and observations.

Please, find the attachement.

Round 2

Reviewer 2 Report

Accepted